# Do restoration strategies in mangroves recover microbial diversity? A case study in the Yucatan peninsula

Daniel Esguerra-Rodríguez[1,2], Arit De León-Lorenzana[2], Claudia Teutli[3], Alejandra Prieto-Davó[4], José Q. García-Maldonado [5], Jorge Herrera-Silveira[5], Luisa I. Falcón [2]*

1 Posgrado en Ciencias del Mar y Limnología, Universidad Nacional Autónoma de México, Ciudad de México, México, 2 Instituto de Ecología, Laboratorio de Ecología Bacteriana, Unidad Mérida, Ucú, Yucatán, México, 3 Escuela Nacional de Estudios Superiores Mérida, Universidad Nacional Autónoma de México, Ucú, Yucatán, México, 4 Facultad de Química, Unidad de Química Sisal, Universidad Nacional Autónoma de México, Sisal, Yucatán, México, 5 Departamento de Recursos del Mar, Centro de Investigación y de Estudios Avanzados del Instituto Politécnico Nacional, Mérida, Yucatán, México

* luisaifalcon@gmail.com, falcon@ecologia.unam.mx

## Abstract

Mangrove forests are fundamental coastal ecosystems for the variety of services they provide, including green-house gas regulation, coastal protection and home to a great biodiversity. Mexico is the fourth country with the largest extension of mangroves of which 60% occurs in the Yucatan Peninsula. Understanding the microbial component of mangrove forests is necessary for their critical roles in biogeochemical cycles, ecosystem health, function and restoration initiatives. Here we study the relation between the microbial community from sediments and the restoration process of mangrove forests, comparing conserved, degraded and restored mangroves along the northern coast of the Yucatan peninsula. Results showed that although each sampling site had a differentiated microbial composition, the taxa belonged predominantly to Proteobacteria (13.2–23.6%), Desulfobacterota (7.6–8.3%) and Chloroflexi (9–15.7%) phyla, and these were similar between rainy and dry seasons. Conserved mangroves showed significantly higher diversity than degraded ones, and restored mangroves recovered their microbial diversity from the degraded state (Dunn test p-value Benjamini-Hochberg adjusted = 0.0034 and 0.0071 respectively). The structure of sediment microbial β-diversity responded significantly to the mangrove conservation status and physicochemical parameters (organic carbon content, redox potential, and salinity). Taxa within Chloroflexota, Desulfobacterota and Thermoplasmatota showed significantly higher abundance in degraded mangrove samples compared to conserved ones. This study can help set a baseline that includes the microbial component in health assessment and restoration strategies of mangrove forests.

## Introduction

Mangroves are tropical and sub-tropical coastal forests that, tolerate salinity, desiccation, flooding, high temperatures, among other challenging conditions associated with the tidal and

**Data Availability Statement:** Sequence reads are available in the NCBI Sequence Read Archive under Bioproject number PRJNA1039151.

**Funding:** DER received a national graduate studies scholarship from CONAHCYT Mexico, No. 788670 as part of his Doctoral degree research in the Posgrado en Ciencias del Mar y Limnologia, UNAM. AdLL received a postgraduate scholarship "Estancias Postdoctorales por Mexico" from CONAHCYT. LIF received funding from UNAM-PAPIIT IN204224.

**Competing interests:** The authors have declared that no competing interests exist.

hydrological regime [1–3]. These forests contribute to the lives and livelihoods of millions of people through ecosystem services that include fishery resources, nurseries, coastal protection and stabilization, acting as carbon sinks [4–6].

Mangrove forests have become one of the most threatened ecosystems due to coastal infrastructure development, industrial activity and urban expansion, factors that cause deforestation and disturbance [7,8]. The increase in intensity and frequency of extreme weather events puts the functionality and conservation of mangrove ecosystems at risk [9,10].

Particularly, microorganisms in mangrove sediments play crucial roles in the biogeochemical cycles, ecosystem health and function [6,11–16]. Bacteria, Archaea and fungi are the main biomass components of mangrove sediments and are key in maintaining mangrove ecosystem services (*e.g* water quality, carbon uptake) [16,17]. Nevertheless, the vegetation structure and abiotic parameters are currently the main elements considered to evaluate the condition and health of mangrove ecosystems [18]. Furthermore, there is little understanding of the response and specific roles of microorganisms in maintaining conserved mangroves, supporting restoration processes and enduring degradation since most studies aim to understand local particular processes that influence microbial patterns [11,19–25]. There are several microbial metabolic routes and interactions that are potentially important for the mangrove's health, that is why it is important to understand the ecological dynamics associated with these microorganisms as a conservation resource [26–29]. The environmental fluctuation, complex forest structure and high productivity, provide unique settings that harbor great biodiversity, which in turn models their microbial component diversity and structure [1,30,31].

In Mexico, mangroves are a significant reservoir of organic carbon (OC) and particularly in the Yucatan Peninsula (YP) they account for >50% of blue carbon storage in sediments. The YP is a karstic platform, with a great diversity of coastal lagoons and heterogeneous environmental conditions that harbor mangroves, and at the same time multiple niches for the microorganisms that inhabit them [32]. The mangroves in the YP have been degraded because of land use changes, infrastructure, urban expansion, pollution, and extreme climatic events. Many of these impacts have been identified and successful rehabilitation programs have been designed for this region [33,34]. Although knowledge of microbial component variation associated to different conditions of mangrove forests in the YP has only started to be incorporated in local projects.

Work in the Red Sea [8] and in Colombia [35] suggest differences in microbial composition in mangroves with higher anthropic impact than in pristine ecosystems, generally showing higher abundance of microbial taxa that potentially respond to disturbance in degraded areas. Further, studies in China highlight that mangrove restoration stimulates microbial diversity recovery in sediments, nutrient availability and extracellular polymeric substance production [36–38]. A recent study of the sediment microbial component in Celestun (northwest YP), reported that the red mangrove, *Rhizophora mangle*, in different ecological types had a more cosmopolitan and heterogenous composition than the less abundant black mangrove, *Avicennia germinans*, that hosts a specific microbiome, suggesting the relevance of incorporating microbial ecology studies to better define restoration strategies of mangrove forests [32]. Hence, the aim of this study was to make a description of the composition of bacteria and archaea from mangrove sediments along the northern YP coast incorporating mangroves in different status: conserved, degraded and restored. We aimed to understand if conserved mangroves shared a particular microbial composition, identify if specific groups of microbes are lost when mangroves become degraded, and which are recovered during the process of restoration. This study sets a baseline of the microbial component found in sediments from mangroves with different conservation status and will impact local and global mangrove restoration initiatives, which are fundamental in tropical regions subject to environmental challenges.

## Materials and methods

### Study site and sample collection

Mangrove ecosystems included in this study are located at the north coast of the YP, Mexico. The study included four locations in the state of Yucatan, from west to east: Sisal, Progreso, Dzilam and Ria Lagartos (Fig 1). Together they cover an area of influence of approximately 430 ha of degraded (without vegetation), restored and conserved mangroves. Hydrological restoration was carried out in all locations hence no reforestation efforts were implemented, and mangrove composition in restored plots are product of natural succession processes. Every

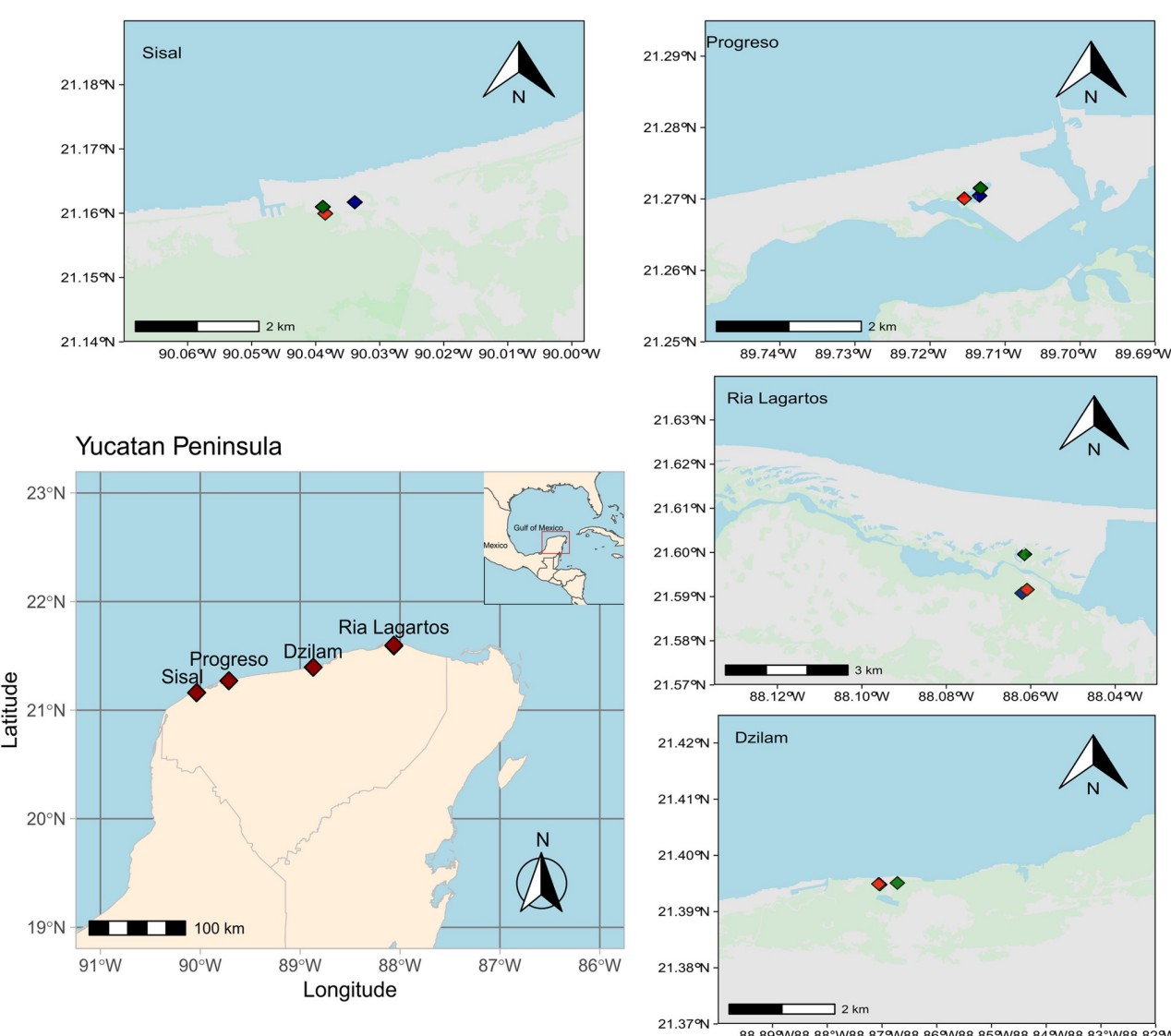

**Fig 1. Study sites and the conservation status of the sampled plots.** In each location red (conserved), green (degraded) and blue (restored) points show the conservation status, light blue layer shows water bodies (INEGI, 2010 [40]) and light green layer shows mangrove coverage (CONABIO, 2021 [41]). The vectorial data of Mexico administrative boundaries are from the following source: https://www.geoboundaries.org. The vectorial data of Yucatan hydrography are from the following source: INEGI, 2010 [40]. The vectorial data of Yucatan's mangrove coverage are from the following source: CONABIO [41]. The pipeline to construct the map with R libraries: *ggplot2*, *ggspatial*, *sf* and *ggpubr* is available in Github (https://doi.org/10.5281/zenodo.11269158). The terms of use of vectorial information can be consulted in https://www.geoboundaries.org/index.html#usage, https://en.www.inegi.org.mx/inegi/terminos.html and http://geoportal.conabio.gob.mx/metadatos/doc/html/mx_man20gw.html respectively [42].

sampling site was associated with coastal lagoon basin mangrove ecosystems, where conserved plots did not share dominant mangrove species with restored plots (Tables 1 and 2). Within each location, plots with different conservation status were sampled (conserved, degraded, restored). Conservation status of the different plots was categorized based on continuous ecosystem monitoring of the forest ecology parameters (diameter at breast height, basal area, density, height, structural complexity index, regeneration potential, productivity) made by "Laboratorio de Produccion Primaria–CINVESTAV" [39], where conserved mangroves had a higher complexity index and density, whereas degraded plots had no vegetation, restored mangroves were those in the area hydrologically restored. Characterization of the sampling sites are shown in Tables 1 and 2 and S1 Table.

Between 8 and 10 samples of superficial sediment (first 10 cm) adjacent to mangrove roots were taken in each plot, during the two main climatic seasons (rainy and dry). Samples were taken with a core sampler (diameter of 5.5 cm), and physical-chemical parameters (salinity, temperature, pH, Redox potential) were measured in interstitial water with portable multi-parametric probes (ATAGO CO. Ltd. hand refractometer and Myron L$^{®}$ Ultrameter II$^{™}$ 6PFCE). Sediments sampled were stored in sterile tubes, refrigerated, and immediately

**Table 1. Description of the sampling sites.**

| Site | Mean Forest density (ind ha$^{-1}$) | Mean flooding level (m) | Mean flooding frequency (events) | Anthropogenic intervention | Start of restoration | Main economic activity |
|---|---|---|---|---|---|---|
| Sisal | 4800 | -0.002 | 0.64 | Intermediate (aquaculture until 2006) | 2009 | Artisanal fishery, tourism |
| Progreso | 4100 | 0.05 | 0.3 | High (extensive urban and touristic impact) | 2011 | Port, tourism |
| Dzilam | 301 | 0.05 | 0.45 | Low | 2009 | Artisanal fishery, tourism |
| Ria Lagartos | 3940 | -0.014 | 0.65 | Low | 2009 | Artisanal fishery, livestock. |

The table summarizes the parameters and characteristics monitored in Herrera-Silveira *et al.*, 2018 [39]

**Table 2. Sampling sites, dominant mangrove species in each conservation status and average physicochemical parameters in interstitial water and sediment.**

| Site | Con_status | Dom_mang_sp | Salinity (psu) | Temperature (˚C) | pH | Redox_pot (V) | OC (%) |
|---|---|---|---|---|---|---|---|
| Ria Lagartos | Conserved | *Avicennia germinans* | 29 (1.8) | 30.9 (0.30) | 7.2 (0.05) | -268 (11.6) | 6.5 (0.11) |
| | Degraded | No vegetation | 13 (0.9) | 32.2 (0.25) | 7.2 (0.03) | -336 (6.3) | 6 (0.35) |
| | Restored | *Laguncularia racemosa* | 37 (1.1) | 31.4 (0.38) | 7.2 (0.03) | -279 (9.2) | 2 (0.13) |
| Dzilam | Conserved | *Avicennia germinans* | 72 (1.5) | 30.9 (0.26) | 6.9 (0.04) | -232 (15.9) | 5.5 (0.45) |
| | Degraded | No vegetation | 60 (3.3) | 32.7 (0.24) | 6.9 (0.03) | -334 (7.6) | 3.4 (0.17) |
| | Restored | *Laguncularia racemosa* | 62 (3.4) | 30.2 (0.35) | 7 (0.03) | -251 (14.1) | 8 (0.25) |
| Progreso | Conserved | *Rhizophora mangle* | 58 (1.6) | 32.6 (0.36) | 7 (0.03) | -285 (5.2) | 5.6 (0.16) |
| | Degraded | No vegetation | 85 (1.9) | 31.8 (0.31) | 7.6 (0.04) | -191 (10.2) | 3 (0.09) |
| | Restored | *Avicennia germinans* | 62 (1.4) | 32.8 (0.49) | 7.3 (0.03) | -288 (5.7) | 3.5 (0.2) |
| Sisal | Conserved | *Avicennia germinans* | 53 (2.2) | 30.8 (0.28) | 6.9 (0.05) | -175 (9.5) | 13 (0.84) |
| | Degraded | No vegetation | 31 (2.3) | 30.9 (0.19) | 7.3 (0.03) | -301 (3.5) | 6.8 (0.1) |
| | Restored | *Laguncularia racemosa* | 27 (1.2) | 31.4 (0.34) | 7.1 (0.06) | -288 (8.6) | 6.4 (0.17) |

The organic carbon (OC) was measured directly from sediments, while the other parameters were measured *in situ* in interstitial water. Values in parenthesis represent the standard error.

transported to the laboratory where they were stored for posterior biogeochemical and molecular analysis. All samples were obtained under national environmental agency permit No. 04857 "Direccion General de Vida Silvestre–Secretaría de Medio Ambiente y Recursos Naturales (SEMARNAT)".

## Sample processing

Sediment microbial composition was characterized from environmental DNA, which was extracted (0.25 g per sample) following the DNeasy PowerSoil Kit (Qiagen, Hilden, Germany) protocol. The DNA was amplified for the 16SrRNA hypervariable V4 region using the 515F-806R primer set, following the protocol described in Gómez-Acata *et al.*, [32]. The amplicon products were purified with magnetic beads (Agencourt AMPure XP, Beckman Coulter) and 20 ng of each sample were sequenced with an Illumina MiSeq (Illumina, San Diego, CA, USA) at the Yale Kegg Center for Genome Analysis. Sequence reads are available in the NCBI Sequence Read Archive under Bioproject number PRJNA1039151.

To determine total carbon, total nitrogen and organic carbon, samples were ground, homogenized and sieved through 250 μm. From each sample 2g were taken to remove inorganic carbon with acidification [43,44]. Treated and untreated samples were subsequently weighed (1g) in silver and tin capsules respectively, and were analyzed with an automatic elemental analyzer (model ECS 4010, Costech Analytica Technologies Inc.) [45]. The parameters for this analysis were: mode CHNS, left reactor between 1020–1050 ˚C, right reactor at 650 ˚C, 2 m chromatography column at 60–80 ˚C, pneumatic autosampler calibrated for 5 min runs. Results were reported in terms of percentage.

## Data analysis

To analyze the sequence reads we used Qiime2 software [46]; briefly, demultiplexed paired sequences were denoised using the DADA2 algorithm [47] with the function qiime dada2 denoise-paired. Based on quality plots, sequences were trimmed at 14 and 234 pair bases for the dry season, and at 14 and 220 for the rainy season. Sequences were clustered in amplicon sequence variants (ASVs) and taxonomy was assigned to variants using the qiime2 function qiime feature-classifier classify-sklearn and SILVA 138-99-nb classifier [48,49].

Phylogeny was constructed based on a mafft alignment [50] and FastTree2 [51] using the qiime2 function qiime phylogeny align-to-tree-mafft-fasttree. Phylogenetic tree, taxonomy table and ASV counts were imported for further analysis in R phyloseq package [52]. Taxa with very low prevalence within the samples were omitted (taxa with less than 2 reads in less than 1% of the total number of samples), synthetic samples, unassigned ASVs at phylum level and Eukarya assignments were also removed.

The filtered dataset was used to estimate alpha diversity as inverse Simpson index (*1/D*) using the phyloseq's function estimate_richness; beta diversity analysis was estimated using vegan package function ordinate [53] from weighted UniFrac distance [54] and visualized in a PCoA [55] with ggplot2 package [56]. A PERMANOVA using the same distance metrics with the adonis function from vegan was used to test for differences between conservation status. An analysis of the most determinant taxa to differentiate sediment microbial composition (differential expression analysis) was done with DESeq2 [57]. For this analysis the factor (conservation condition) size effect was estimated using geometric means of the counts, and the dispersions were calculated on locally fitted data to negative binomial distribution with maximum likelihood estimates, with its results expressed as log fold change. The Approximate Posterior Estimation for generalized linear model was used to moderate the logarithmic fold change estimated as the differential expression [58]. The physical-chemical parameters and

organic carbon estimations were incorporated along the conservation status into a canonical correspondence analysis (CCA) [55] to describe possible predictors to explain differences in composition between samples. The pipeline to the raw sequence analysis in Qiime2 and for the exploration and further statistical analysis in R is available in Github (https://doi.org/10.5281/zenodo.11122245).

## Results

### Sediment microbial diversity and composition

The number of sequences (single reads) obtained for the dry and rainy seasons were 12,865,108 and 14,099,658 respectively. After alignment and denoising 47,570 and 55,098 representative sequences were recovered for dry and rainy seasons correspondingly. A total of 210 samples and 76,369 ASVs were recovered for analysis, corresponding to four locations in the YP, including three mangrove conservation conditions, during rainy and dry seasons.

The alpha diversity and taxonomic composition analyses suggested that sediment microbial composition of mangroves differs between conservation status. The samples with the highest diversity (*1/D*) corresponded to conserved and restored conditions (Fig 2) (Kruskall—Wallis test: p-value = 0.004, chi-squared = 10.865). The highest differences between mangrove conditions were found between degraded and restored (Dunn test: Z = -2.59, p-value = 0.007) and degraded versus conserved mangroves (Dunn test: Z = 3.05, p-value = 0.003). Sisal mangroves were an exception since degraded and conserved conditions showed higher microbial diversity than restored plots (Fig 2). In Sisal, the degraded site receives marine water influx.

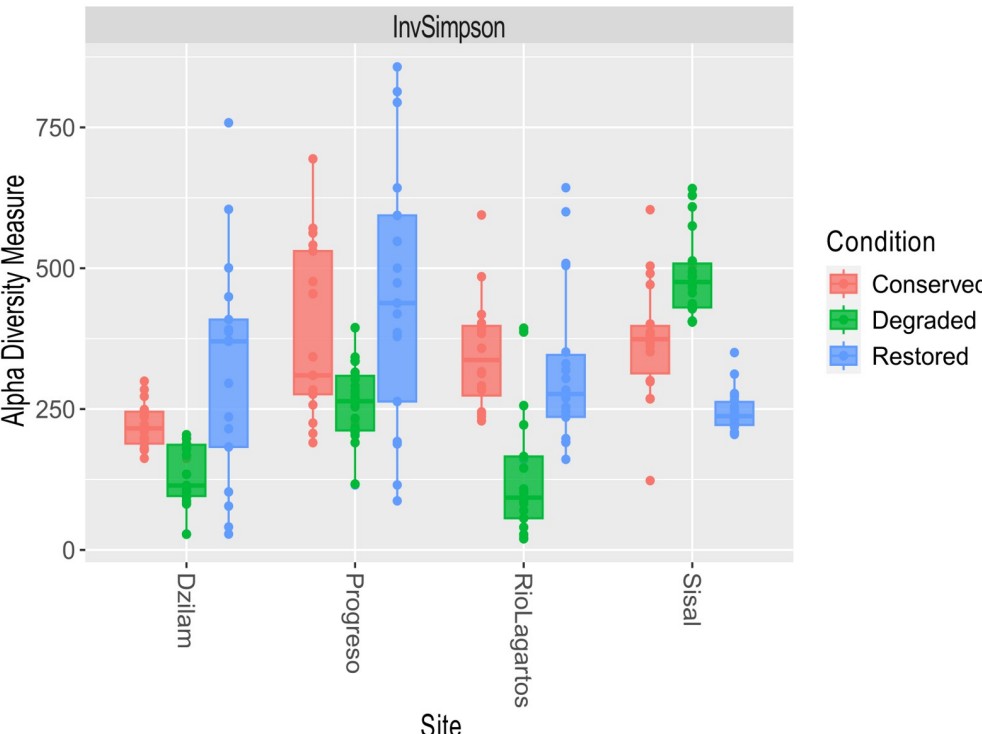

**Fig 2. Alpha diversity (*1/D*).** Microbial genetic diversity (inverted Simpson index) in sediments along the northern coast of Yucatan.

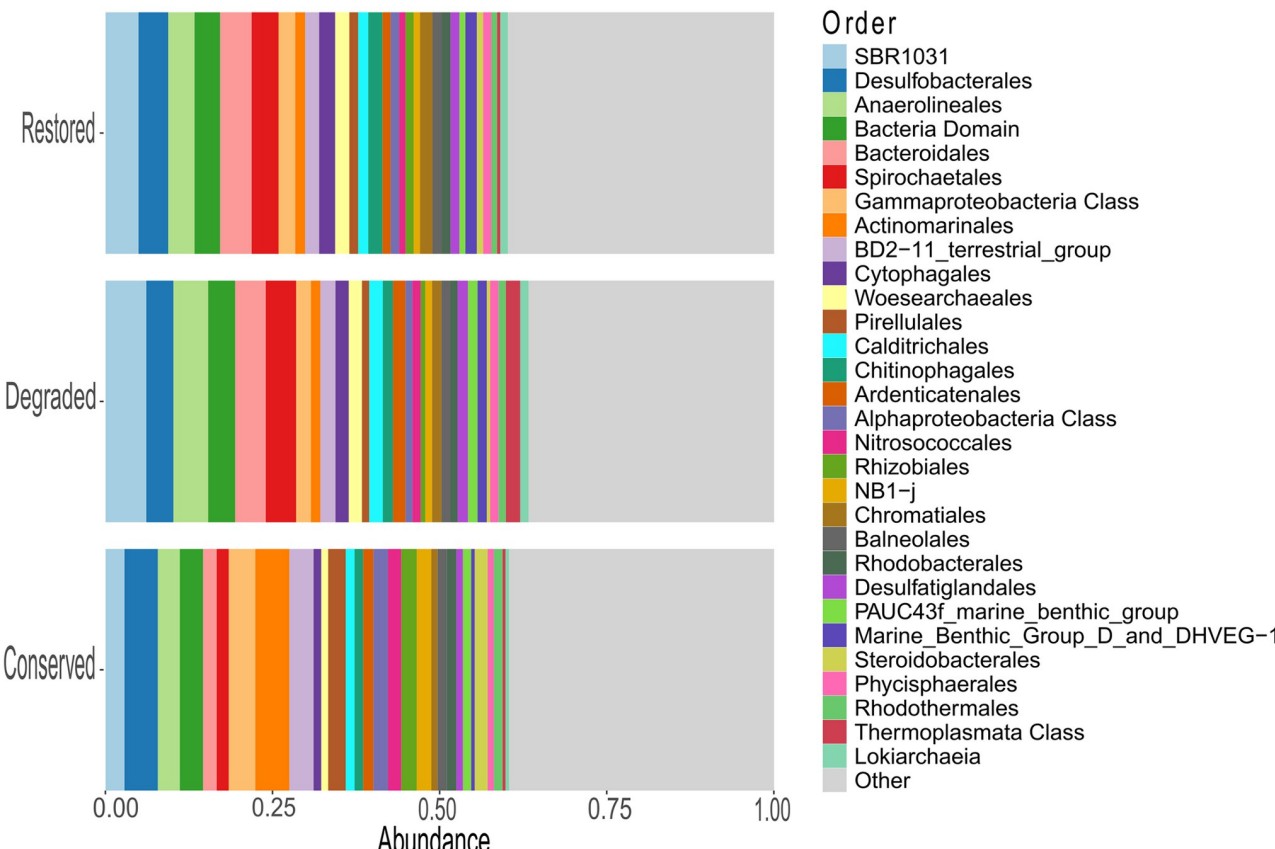

**Fig 3. Relative microbial abundance.** The most abundant microbial orders found along the northern coast of Yucatan. Taxa within the "Other" category have abundances lower than 0.5%.

Overall, the most abundant phyla were Proteobacteria (13.2–23.6%), Desulfobacterota (7.6–8.3%), Chloroflexi (9–15.7%), Bacteroidota (8.3–15.1%), Planctomycetota (5.9–8%), Gemmatimonadota (3.3–5.7%), Actinobacteriota (2–6.6%), Crenarchaeota (1.1–2.1%), Acidobacteriota (2.6–4.9%), Spirochaetota (1.9–4.7%), Nanoarchaeota (1–2.1%), Cyanobacteria (0.6–1.8%), Myxococcota (1.5–1.9%), Verrucomicrobiota (1–1.9%) and Thermoplasmatota (1–4,4%) (S1 Fig). At lower taxonomic levels, orders including Rhizobiales (2.3%) and Actinomarinales (5.1%) showed higher (two times or more) average abundance in conserved than the other conservation status. Other taxa including SBR1031 (6.1%) and "Unassigned Order Thermoplasmata Class" (2.1%) showed greater (two times or more) average abundance in degraded than conserved samples (Fig 3). Mangrove sediments in Dzilam and Sisal showed Desulfobacterales, Desulfatiglandales and Bacteroidales in greater proportions in degraded mangroves (S2 Fig).

## Mangrove forest conservation status and microbial structure

Mangrove conservation status and location are the main factors that explain the microbial composition of sediments ($R^2$ condition = 0.0769, F condition = 11.075, $R^2$ site = 0.2145, F site = 20.592, p-values = < 0.001). The differences between sediment from degraded and conserved mangroves are more evident in Sisal and Dzilam locations, while degraded and restored samples are closely aggregated (Fig 4). Ria Lagartos sediments show the smallest dispersion

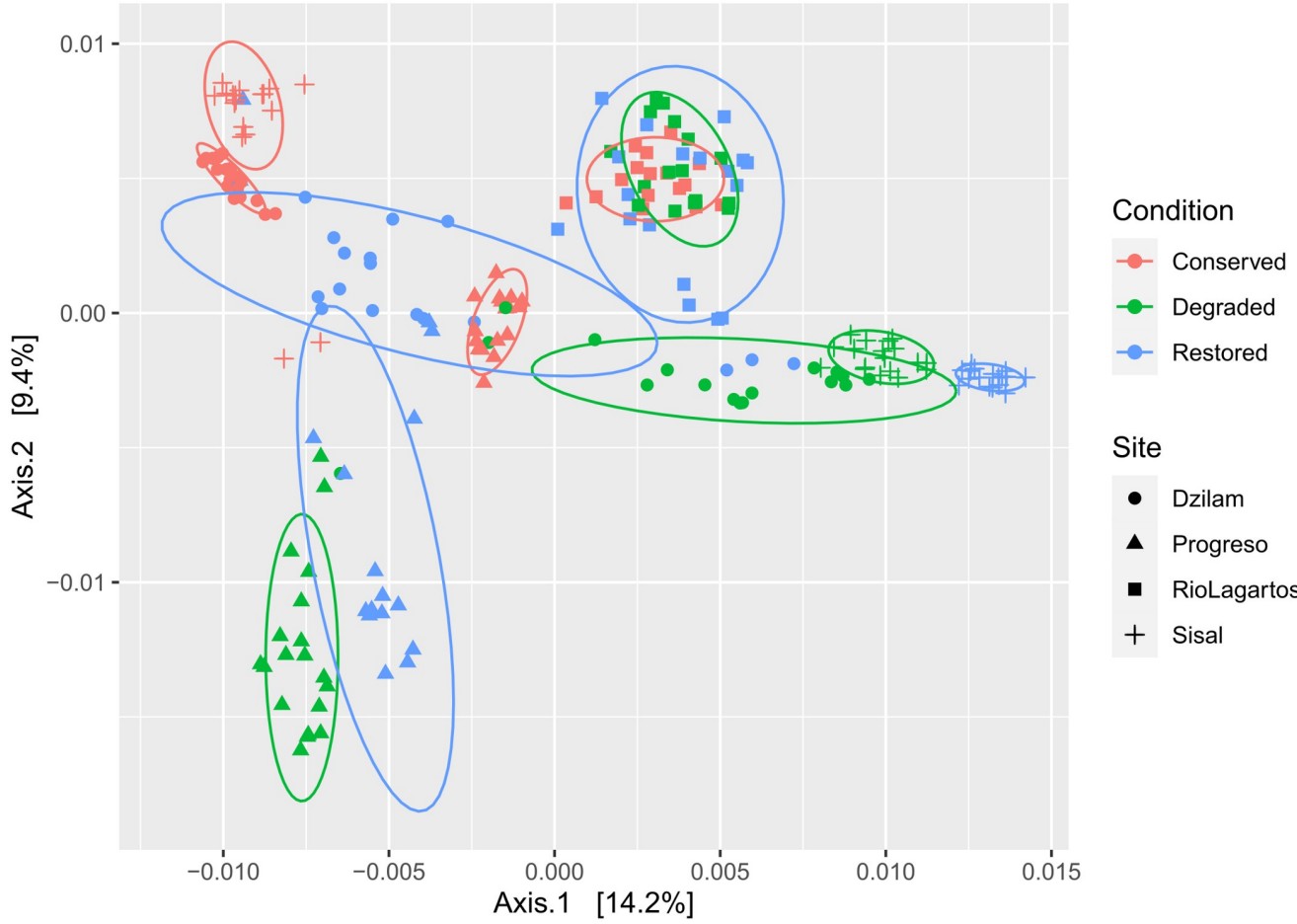

**Fig 4. Principal coordinate analysis (PCoA).** Dispersion of sediment microbial composition along the northern Yucatan coast. Distances were measured with Weighted UniFrac metrics. Ellipses account for 85% of the samples within each group. Mangrove conservation status (colors) and study sites (shapes).

and distances between the different conservation status. Samples from degraded and restored mangroves from Progreso are grouped apart from the rest (Fig 4). Additionally, pairwise comparisons performed between samples according to their conservation status for each location determined significant differences in all cases (S2 Table).

Canonical correspondence analysis (CCA) for each location confirms that sediments from different conservation status ordinate separately, in this analysis differences between conservation status seem more contrasting in Sisal and Progreso. The ordination of sediment microbial diversity also responded significantly to organic carbon content, redox potential, and salinity (Fig 5 and S3 Table).

## Microbial differential abundance

Sediment microbial composition from mangrove forests along the YP coast have common patterns despite their conservation status, and prevalent taxa that may constitute a core microbiome (S3 Fig, S4 Table). Nonetheless, there are taxa that show significant differences in abundances between conserved and degraded mangroves as shown in an analysis of differential expression that identified 951 ASVs (lowest Benjamini-Hochberg adjusted p-

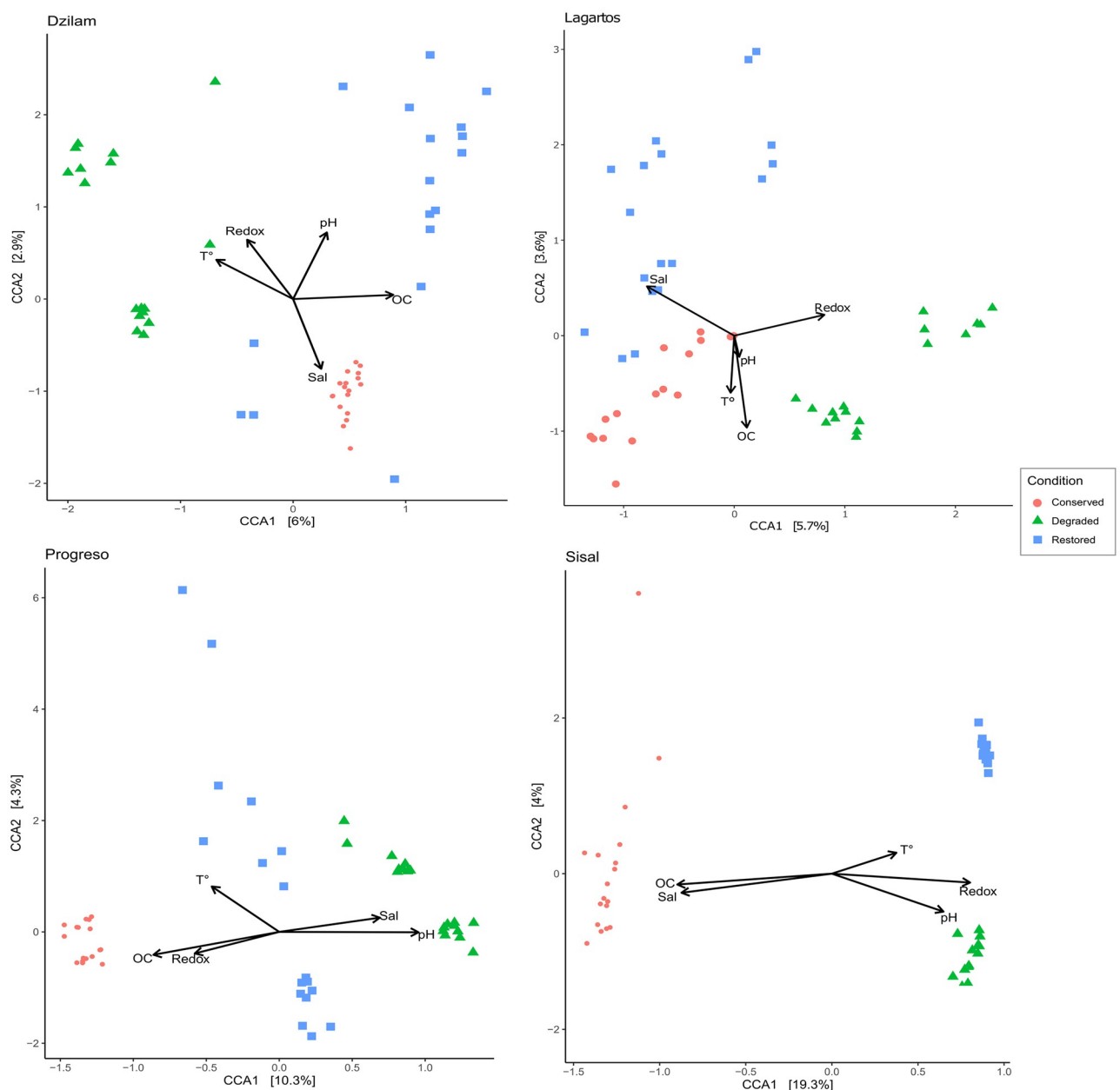

**Fig 5. Canonical correspondence analysis (CCA).** Ordination from weighted UniFrac distances. Conservation status: red circles (conserved), green triangles (degraded), blue squares (restored). Vectors correspond to physicochemical variables: temperature (T°), salinity (Sal), organic carbon (OC), Redox potential and pH.

values < 0.01) (Fig 6). When comparing conserved and restored samples, 213 ASVs showed significant differential abundances (not shown), while 99 ASVs showed significant differential abundances when degraded and restored samples were compared (Fig 7).

Differences displayed between conservation status were in many cases related to the orders and phyla previously mentioned as the most abundant throughout the study. There are taxa with the potential to be indicators of degraded mangrove sediments, including anaerobic groups like Chloroflexota orders: SBR1031 and Anaerolineales. Desulfobacterota orders like

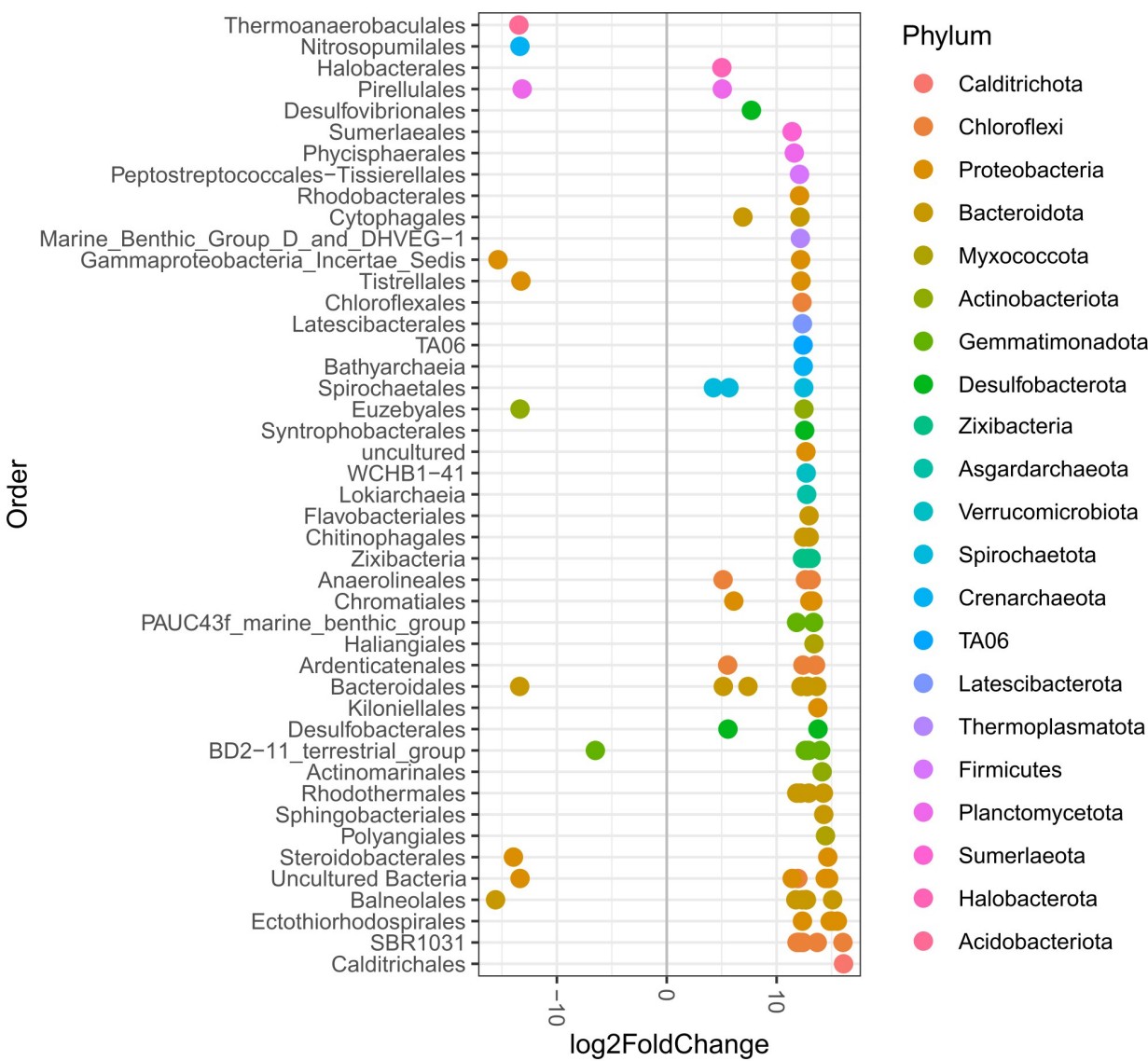

**Fig 6. ASVs (including all sites) that show significant differences in their abundance (between degraded and conserved mangroves).** Orders (vertical axis) and corresponding phyla (colors) along the northern YP coast are shown. Positive values show more abundant ASVs in degraded samples, while negative values show less abundant ASVs in degraded samples.

Desulfobacterales and Desulfatiglandales had higher abundances in degraded samples, also halophilic and thermophilic archaea including Thermoplasmatota and Halobacterota groups significantly differentiated sediments from conserved mangroves compared to degraded mangroves across the different study sites (S5 Table).

Each site harbored differential abundances between conservation status. Ria Lagartos had 147 ASVs that were significantly different between conserved and degraded plots, including Desulfobulbales, Desulfatiglandales, Lachnospirales, OPB41, Chlorobiales, WCHB1-81, Desulfobacterales and Steroidobacterales (S4A Fig and S6 Table). While Nitrosococcales, Cytophagales, Rhodobacterales, and several orders of Desulfobacterota, among others, showed higher abundances in restored than in degraded mangroves (S4B Fig and S7 Table).

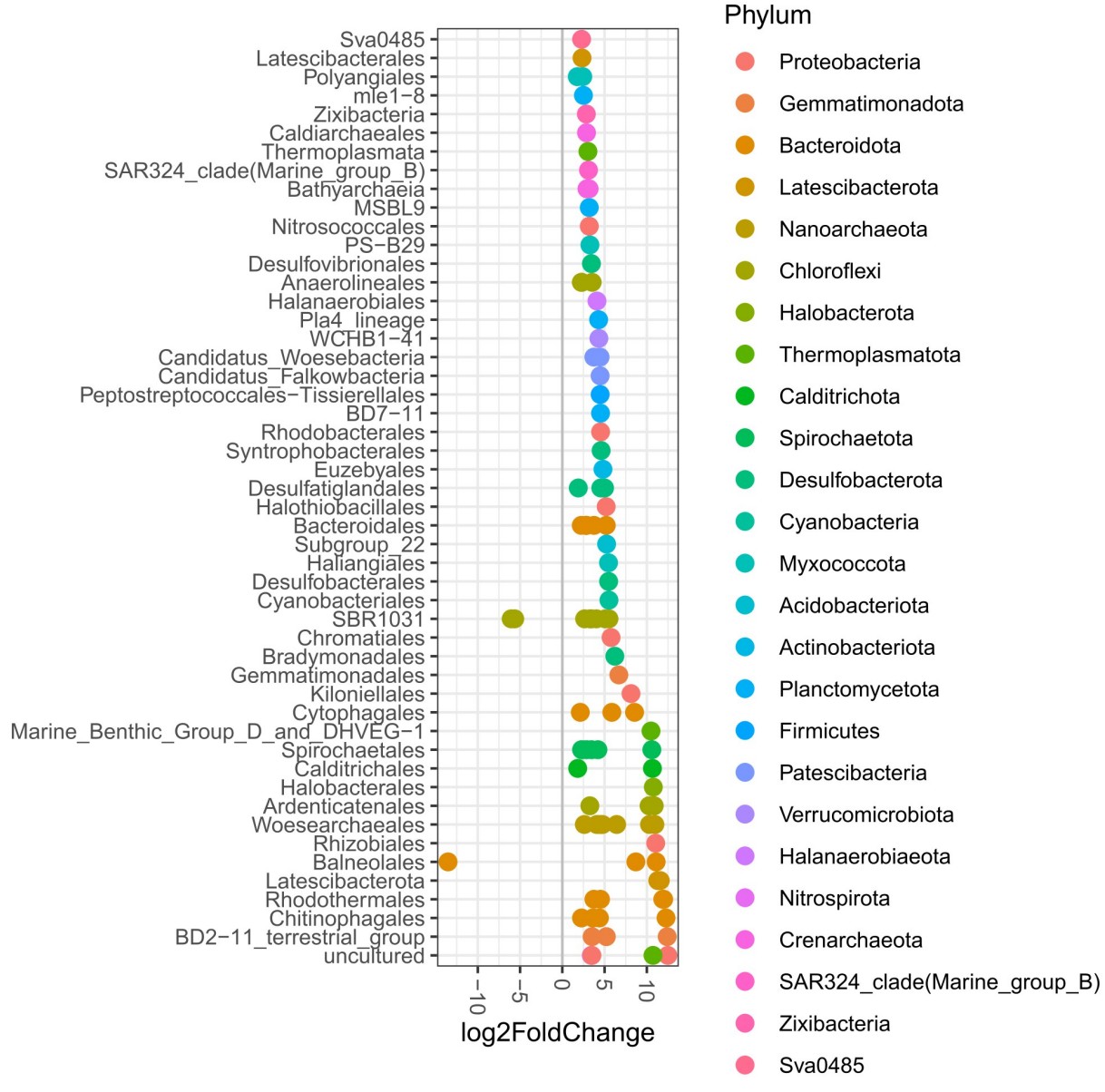

**Fig 7. ASVs (including all sites) that show significant differences in their abundance (between degraded and restored mangroves).** Orders (vertical axis) and corresponding phyla (colors) along the northern YP are shown. Positive values show more abundant ASVs in degraded samples, while negative values show less abundant ASVs in degraded samples.

In the case of Dzilam, 600 ASVs had significant differences in abundance between conserved and degraded status including PAUC43f, Ardenicatenales, Euzebyales, Polyangiales, Subgroup 26 (Acidobacteriota), Thermoanaerobaculales, Pirellulales, Actinomarinales, Balneolales, Vicinamibacterales, BD2–11 terrestrial group, NB1–j, Subgroup 17 (Acidobacteriota), Terrybacterales, Rhodobacterales, Subgroup 21 (Acidobacteriota), Rhodothermales, Microtrichales and Nitrososphaerales (S5A Fig and S8 Table). Restored mangrove sediments were significantly more abundant than degraded sediments in Actinomarinales, Nitrosococcales, Nitrosopumilales, Balneolales, Cytophagales, Rhodothermales, Rhizobiales, as well as

other groups of Proteobacteria, Bacteroidota, Acidobacteriota and Gemmatimonadota (S5B Fig and S9 Table).

The mangroves of Progreso had 950 ASVs representing differential abundances when comparing conserved and degraded mangroves. Conserved mangroves had more abundant taxa in the orders Actinomarinales, Subgroup 17 (Acidobacteriota), Latescibacterales, Sphingomonadales, Syntrophobacterales, Rhodothermales, Desulfobulbales, PAUC43f, Steroidobacterales, Bathyarchaeia, Subgroup 21 (Acidobacteriota), Cellvibrionales and Vicinamibacterales (S6A Fig and S10 Table). Restored mangroves were more abundant than degraded ones in Spirochaetales, Kiloniellales, MBG-D_and_DHVEG-1, Nitrosopumilales, Phycisphaerales, Latescibacterales, Aminicenantales, Sphingobacteriales, and groups in the classes Bathyarchaeia and Lokiarchaeia (S6B Fig and S11 Table).

In Sisal, 1562 ASVs with differential abundance between conserved and degraded conditions were identified, where conserved mangrove sediments were significantly more abundant in Nitrospirales, Nitrosococcales, Tistrellales and Balneolales (S7A Fig and S12 Table). Restored mangroves sediment samples were significantly more abundant than degraded samples in Ignavibaterilaes, Rhizobiales, Synechococcales, Rhodobacterales, LCP-89, Nannocystales, Desulfomonilales, Silvanigrellales, Cyanobacteriales, Flavobacteriales, Aminicenantales, groups within the classes Odinarchaeia and Lokiarchaeia, and groups within the phylum Thermoplasmatota (S7B Fig and S13 Table).

## Discussion

Mangrove forests in the YP have an extension of over 500,000 ha constituting 60% of mangroves in Mexico [59]. The most abundant mangrove species in the YP are *Rhizophora mangle* (red mangrove), *Avicennia germinans* (black mangrove), *Laguncularia racemosa* (white mangrove)) and *Conocarpus erectus* (buttonwood) [59]. Previous studies in recently restored mangrove forests in the YP suggest that *L. racemosa* is a pioneer species since it has the highest growth rates [60]. Here, three of the four study areas had dominance of *L. racemosa* in the restored plots except in Progreso where *A. germinans* dominated. Conserved mangroves had dominance of *A. germinans* except again in Progreso where *R. mangle* was dominant. All sites analyzed had a differential mangrove composition between conserved and restored plots, yet mangrove species was not a variable that grouped sediment microbial composition along different study sites. Although, the results of this study strongly suggest that microbial composition responds to mangrove conservation status, further research that explores specific mangrove species relation with microbial composition is still needed. In this regard, a study in mangroves from the northwest YP [32] highlights that *R. mangle* associated microbial diversity is more generalist compared to *A. germinans*.

The indicator parameters that address the ecosystem condition are those related to the vegetation, sediments, and water [61]. Within this framework, conserved mangroves of the YP have been typically described with high foliage coverage and productivity, higher establishment of seedlings, and high environmental heterogeneity that facilitate resources and hydroperiod gradients. The inverse pattern is depicted for degraded areas, in which no mangrove trees are established. Restored mangroves can show intermediate conditions between these two, depending on the success and stage of the restoration process [62–64]. Restoration strategies included in this study are hydrological, hence there is no reforestation and mangrove forests are naturally recovering their coverage [39]. There are complex processes that lead to mangrove forest recovery, and in this study, patterns associated to conservation status were found to be site-specific. The response of microbes to degradation, restoration or conserved environments may help discern the ecological functioning of different mangroves and give

hints to better address the managing of these ecosystems and improve restoration policies, especially in mangroves with high potential for climate change mitigation such as those in the YP [65].

## Microbial response to mangrove conservation status

Mangrove forests have sediment microbial communities that play crucial roles in the biogeo-chemical cycles of the ecosystem. In this study, bacterial and archaeal diversity from mangrove sediments show changes depending on the conservation status of the mangrove forest and therefore may be indicators of ecosystem health. Mangrove sediments along the northern YP show phyla abundance similarities with other mangrove ecosystems [16,66–76]. Also, differences between the microbial communities in mangroves with contrasting conservation status are associated to local sediment environmental conditions including salinity, redox potential, and organic carbon content. Specific taxa are commonly found in mangroves with the same conservation status along the YP coast, for example: Chloroflexi, Bacteroidota, Proteobacteria and Actinobacteria phyla were always found in conserved mangroves. Furthermore, no significant differences were observed between restored and conserved mangrove microbial diversity, leading us to think that restoration efforts are working towards recovering sediment microbial functionality, however this hypothesis needs to be further tested using other approaches like comparative metagenomics [77].

Ria Lagartos is a Natural Biosphere Reserve and as such, subject to management and protection [78]. The results from this site show the relevance of ecosystem management, where although sediment microbial diversity is significantly different among conservation status there is a smaller dispersion, which suggests a more resilient ecosystem that shows more homogeneous conditions and microbial composition. In general, conserved mangroves had higher values of alpha diversity, except those from Sisal where degraded plots were the most diverse. These plots are located near the mouth of the swamp where hydrodynamics favors the flux with the sea, and at these small-scale physical characteristics can also promote the constant migration of microbial pools [79,80]. Richness is known to be an important factor to ecosystems functionality [81,82] and in this study, high microbial diversity in sediments was an indicator in most of the conserved mangroves, highlighting the importance of these microbial communities in maintaining ecosystem functions in conserved regions. However, high richness has also been associated with polluted or anthropogenically impacted systems [83], where input of organic and inorganic compounds has been hypothesized to give a variety of substrates for many opportunistic microbial species that can use them [76]. The above might be the case in Sisal, where higher alpha diversity values were associated with the degraded mangroves and organic carbon content, presumably linked to historically reported aquaculture and urban discharges [84].

The YP is characterized for being a karstic platform whose geomorphology depends on the dissolution of carbonate minerals. A characteristic formation in the region are sinkholes (cenotes, from the mayan *dzonot*), which are part of a complex underground freshwater system [85]. Dzilam, Sisal and Ria Lagartos are located in the area of influence of the ring of cenotes, where greater permeability and underground freshwater discharge depict conditions with lower salinity and small temperature fluctuations in interstitial water [85]. The environmental stability of these locations (especially in Ria Lagartos and Dzilam) may allow the mangroves to compromise less in homeostatic maintenance and have faster growth rates, more resources allocated in tissue formation, hence more organic matter input to the sediments [86]. The microbial communities of the sediments in these locations are more similar even if they were linked to mangrove plots with a different conservation status. In these mangroves, the

microbial diversity and its functional redundancy may help maintain the biogeochemical functions of the sediment and confer resilience to the ecosystem [87,88]. Studies in the northern Yucatan coast evidence that restoration actions result in a faster recovery of mangroves' health, particularly in locations like Ria Lagartos and Dzilam [39,89].

Progreso is an example of environments in the northern coast of the YP with more arid conditions, due to its weathering processes this site has lower permeability than other subregions [85]. Furthermore, this location has had higher anthropogenic intervention resulting in hydrologic modifications, which in turn have changed mangrove coverage and structure [39]. There is a significant difference in abundance between degraded and conserved mangroves in taxa within Gemmatimonadota (BD2-11 terrestrial group, PAUC43F marine benthic group and MD2902-B12), which have been reported in high relative abundance in arid and marine conditions [90–92].

The restored mangroves depend in their early stages mainly on allochthonous sources of carbon and nutrients [63], where mangroves with longer times in the restoration process should therefore have microbial communities that are more similar to those in the conserved reference mangrove. Nonetheless, in the Yucatan karstic system the input coming from underground freshwater could also bring high concentrations of compounds that can be used as substrate or electron acceptors by diverse microbes, helping restore microbial geochemical functionality in sediments faster than in locations with mostly autochthonous sources of nutrients and carbon. Although no significant differences were found between climatic seasons and sediment microbial composition, the effect of hydroperiod variability and underground water inputs should be further studied in different time scales. Furthermore, urban expansion will probably increase, along with pollution from sewage coming through the underground freshwater system, and if not managed correctly it might affect the dynamics of mangroves of northern YP and its associated microbes [35].

## Potential of microbes to monitor restoration

In this study, physicochemical parameters and organic carbon associated significantly to sediment microbial diversity. Particular microbial groups can create a resilient environment, maintaining a basal biogeochemical functionality that helps the survival of the vegetation and facilitates seedling recruitment [93–95]. Furthermore, microbial taxa can serve as an indicator of degradation processes. In this regard the presence of taxa like Sphingobacteriales, BD2-11-terrestrial-group and SBR1031 that in this study are differentially abundant in contrasting conservation status is remarkable, since they have been reported in maintaining a priming effect when allochthonous organic carbon sources are available in different terrestrial soils [96]. Taxa that represented differences between conservation status in this study, ASVs within Bacteroidota and Gemmatimonadota, have been reported associated with high salinities in China coastal estuarine wetlands [80]. Other taxa, within Acidobacteriota and Chloroflexi were related to low salinities in the same study, and showed ASVs that were more abundant in conserved mangroves from Ria Lagartos, Dzilam and Progreso. Taxa within Acidobacteriota and Chloroflexi could be further explored as indicators of specific biogeochemical activity in mangrove ecosystems. Nonetheless, proposed assemblages would not necessarily be indicative of the same processes or conditions in other ecological types of mangroves (e.g., fringe mangroves with direct marine influence) or in mangrove ecosystems composed by other mangrove species, since microbial communities' composition has been suggested to be species-specific [32,97].

Conserved and restored mangrove sediments are dominated by the rhizobiome of the dominant vegetal species which in this study included *L. racemosa*, *A. germinans* and *R.*

*mangle*. *L. racemosa* was the dominant species in hydrologically restored plots. Potentially, the microbial component associated to the rhizobiome could facilitated the succession process to more stable conditions where *A. germinans* dominates (e.g. Ria Lagartos, Dzilam, Sisal). Prevalence of ASVs that could be associated to specific mangrove species biogeochemical requirements, included taxa within Acidobacteriota, Rhizobiales, Nitrosopumilales and Nitrosococcales, that were enriched in conserved and restored conditions. Representatives of these and other groups were prevalent through this study and have been related with nitrification and other biogeochemical processes important for mangroves, including mineralization of carbon and sulfate reduction [98–101]. These microbial taxa should be further studied to understand their functional role in mangroves subject to hydrologic restoration [102]. Therefore, this study contributes to understanding the delicate and complex processes of hydrological restoration in mangrove forests, where knowledge of the microbial component is important to create well-constructed baselines and promote ecosystem resilience.

## Conclusions

The mangroves of the northern coast of the YP are complex ecosystems where different mangrove species dominate depending on multiple factors. *L. racemosa* has been suggested to be a pioneer species after hydrological restoration, and in this study, dominated three of four restored sites. *A. germinans* was the dominant species in Progreso after restoration and in the rest of the sites in conserved plots. Previous studies have suggested that *A. germinans* has a specific rhizobiome and here, it was evident that this composition is also associated to each mangrove location, shedding new light to the complex patterns of rhizobiome-mangrove associations. Degraded plots harbor a microbial composition that allows biogeochemical functionality which when enhanced by hydrological restoration practices, modifies the conditions to promote the establishment of mangroves. Further understanding of the complexity of the dynamics that lead to the establishment of the rhizobiome is needed for each mangrove species and presumably, for each location. These processes help assure the survival of the vegetation and endure stress, facilitating seedling recruitment. This study shows that sediment microbial communities have great potential to be an integral part in the evaluation and management of mangrove ecosystems and should be explored further specially to understand their functional role in mangrove forest health and biogeochemical cycling.

## Supporting information

**S1 Fig. Bar plot of the most abundant phyla in the four study sites.** Relative abundance (y-axis). Bottom labels are the samples and their corresponding climatic season.
(TIF)

**S2 Fig. Heatmap of most abundant bacterial/archaeal orders.** Relative abundances arranged by mangrove conservation condition (horizontal axis upper label) and the four study sites (horizontal axis lower label).
(TIF)

**S3 Fig. Venn diagram.** Shows the core microbiome (50% prevalence in the group pictured, reads with abundances above 0.1%) along the conservation status. The values inside the ellipses are the number of core ASVs and in parentheses the average relative abundance.
(TIF)

**S4 Fig. The one hundred most significantly different ASVs (orders) between the conservation status in Ria Lagartos.** Log2 fold change comparing differentially abundant taxa (positive

values are ASVs that have significantly higher abundance than degraded samples, while negative values are ASVs that are significantly less abundant than degraded samples). (A) Compares Conserved vs Degraded. (B) Compares Restored vs Degraded.
(TIF)

**S5 Fig. The one hundred most significantly different ASVs (orders) between the conservation status in Dzilam.** Log2 fold change comparing differentially abundant taxa (positive values are ASVs that have significantly higher abundance than degraded samples, while negative values are ASVs significantly less abundant than degraded samples). (A) Compares Conserved vs Degraded. (B) Compares Restored vs Degraded.
(TIF)

**S6 Fig. The one hundred most significantly different ASVs (orders) between the conservation status in Progreso.** Log2 fold change comparing differentially abundant taxa (positive values are ASVs significantly less abundant than degraded samples, while negative values are ASVs significantly less abundant than degraded samples). (A) Compares Conserved vs Degraded. (B) Compares Restored vs Degraded.
(TIF)

**S7 Fig. The one hundred most significantly different ASV's (orders) between the conservation status in Sisal.** Log2 fold change comparing differentially abundant taxa (positive values are ASVs that have significantly higher abundance than degraded samples, while negative values are ASVs significantly less abundant than degraded samples). (A) Compares Conserved vs Degraded. (B) Compares Restored vs Degraded.
(TIF)

**S1 Table. Raw physicochemical information associated to the sediment samples.**
(TXT)

**S2 Table. PERMANOVA pairwise comparisons between conservation status, for each study site.**
(PDF)

**S3 Table. PERMANOVA results of the model used in the CCA.** Shows the values of the multiple physical-chemical parameters that cause the divergence between the conservation status.
(PDF)

**S4 Table. Core microbiome taxa list.**
(CSV)

**S5 Table. ASVs with significant (adjusted p-value < 0.01) abundance differences between degraded and conserved status along all the study sites.** The reference point of comparison is the conserved status. The column "baseMean" are the normalized mean abundance of the ASV, "log2FoldChange" is the logarithmic abundance difference between the compared status, "lfcSE" is the standard error on the logarithmic scale, "padj" is the Benjamini-Hochberg adjusted p-value.
(TXT)

**S6 Table. ASVs with significant (adjusted p-value < 0.01) abundance differences between degraded and conserved status in Ria Lagartos.** The column "baseMean" are the normalized mean abundance of the ASV, "log2FoldChange" is the logarithmic abundance difference between the compared status, "lfcSE" is the standard error on the logarithmic scale, "padj" is

the Benjamini-Hochberg adjusted p-value.
(TXT)

**S7 Table. ASVs with significant (adjusted p-value < 0.01) abundance differences between degraded and restored status in Ria Lagartos.** The column "baseMean" are the normalized mean abundance of the ASV, "log2FoldChange" is the logarithmic abundance difference between the compared status, "lfcSE" is the standard error on the logarithmic scale, "padj" is the Benjamini-Hochberg adjusted p-value.
(TXT)

**S8 Table. ASVs with significant (adjusted p-value < 0.01) abundance differences between degraded and conserved status in Dzilam.** The column "baseMean" are the normalized mean abundance of the ASV, "log2FoldChange" is the logarithmic abundance difference between the compared status, "lfcSE" is the standard error on the logarithmic scale, "padj" is the Benjamini-Hochberg adjusted p-value.
(TXT)

**S9 Table. ASVs with significant (adjusted p-value < 0.01) abundance differences between degraded and restored status in Dzilam.** The column "baseMean" are the normalized mean abundance of the ASV, "log2FoldChange" is the logarithmic abundance difference between the compared status, "lfcSE" is the standard error on the logarithmic scale, "padj" is the Benjamini-Hochberg adjusted p-value.
(TXT)

**S10 Table. ASVs with significant (adjusted p-value < 0.01) abundance differences between degraded and conserved status in Progreso.** The column "baseMean" are the normalized mean abundance of the ASV, "log2FoldChange" is the logarithmic abundance difference between the compared status, "lfcSE" is the standard error on the logarithmic scale, "padj" is the Benjamini-Hochberg adjusted p-value.
(TXT)

**S11 Table. ASVs with significant (adjusted p-value < 0.01) abundance differences between degraded and restored status in Progreso.** The column "baseMean" are the normalized mean abundance of the ASV, "log2FoldChange" is the logarithmic abundance difference between the compared status, "lfcSE" is the standard error on the logarithmic scale, "padj" is the Benjamini-Hochberg adjusted p-value.
(TXT)

**S12 Table. ASVs with significant (adjusted p-value < 0.01) abundance differences between degraded and conserved status in Sisal.** The column "baseMean" are the normalized mean abundance of the ASV, "log2FoldChange" is the logarithmic abundance difference between the compared status, "lfcSE" is the standard error on the logarithmic scale, "padj" is the Benjamini-Hochberg adjusted p-value.
(TXT)

**S13 Table. ASVs with significant (adjusted p-value < 0.01) abundance differences between degraded and restored status in Sisal.** The column "baseMean" are the normalized mean abundance of the ASV, "log2FoldChange" is the logarithmic abundance difference between the compared status, "lfcSE" is the standard error on the logarithmic scale, "padj" is the Benjamini-Hochberg adjusted p-value.
(TXT)

## Acknowledgments

DER is grateful to Posgrado en Ciencias del Mar y Limnología, UNAM, and to CONAHCyT for a graduate studies scholarship. The authors are grateful to Dr. Osiris Gaona (IE, UNAM), MSc. Korynthia Lopez (FQ, UNAM), Dr. Joanna Ortiz (FC, UNAM) and Jessica Olivares (FQ, UNAM) for their technical assistance.

## Author Contributions

**Conceptualization:** Daniel Esguerra-Rodríguez, Claudia Teutli, Alejandra Prieto-Davó, José Q. García-Maldonado, Jorge Herrera-Silveira, Luisa I. Falcón.

**Data curation:** Daniel Esguerra-Rodríguez, Arit De León-Lorenzana, Claudia Teutli, Luisa I. Falcón.

**Formal analysis:** Daniel Esguerra-Rodríguez, Arit De León-Lorenzana, Claudia Teutli, Luisa I. Falcón.

**Funding acquisition:** Claudia Teutli, Jorge Herrera-Silveira, Luisa I. Falcón.

**Investigation:** Daniel Esguerra-Rodríguez, Arit De León-Lorenzana, Claudia Teutli, José Q. García-Maldonado, Jorge Herrera-Silveira, Luisa I. Falcón.

**Methodology:** Daniel Esguerra-Rodríguez, Arit De León-Lorenzana, Claudia Teutli, José Q. García-Maldonado, Jorge Herrera-Silveira, Luisa I. Falcón.

**Project administration:** Jorge Herrera-Silveira, Luisa I. Falcón.

**Resources:** Jorge Herrera-Silveira, Luisa I. Falcón.

**Software:** Daniel Esguerra-Rodríguez.

**Supervision:** Arit De León-Lorenzana, Alejandra Prieto-Davó, José Q. García-Maldonado, Jorge Herrera-Silveira, Luisa I. Falcón.

**Validation:** Daniel Esguerra-Rodríguez, Alejandra Prieto-Davó, Jorge Herrera-Silveira, Luisa I. Falcón.

**Visualization:** Daniel Esguerra-Rodríguez, Alejandra Prieto-Davó, Luisa I. Falcón.

**Writing – original draft:** Daniel Esguerra-Rodríguez, Arit De León-Lorenzana, Alejandra Prieto-Davó, José Q. García-Maldonado, Jorge Herrera-Silveira, Luisa I. Falcón.

**Writing – review & editing:** Daniel Esguerra-Rodríguez, Arit De León-Lorenzana, Claudia Teutli, Alejandra Prieto-Davó, José Q. García-Maldonado, Jorge Herrera-Silveira, Luisa I. Falcón.

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
