## [Decision Letter · Decision Letter 0]

27 Mar 2024

PONE-D-24-03277Do restoration strategies in mangroves recover microbial diversity? A case study in the Yucatan peninsulaPLOS ONE

Dear Dr. Falcón,

Thank you for submitting your manuscript to PLOS ONE. After careful consideration, we feel that it has merit but does not fully meet PLOS ONE’s publication criteria as it currently stands. Therefore, we invite you to submit a revised version of the manuscript that addresses the points raised during the review process.

We look forward to receiving your revised manuscript.

Kind regards,

Jiang-Shiou Hwang, Ph.D.

Academic Editor

PLOS ONE

“DER received a national graduate studies scholarship from CONAHCYT, Mexico as part of his Doctoral degree research in the Posgrado en Ciencias del Mar y Limnologia. AdLL received a postgraduate scholarship “Estancias Postdoctorales por Mexico” from CONAHCYT. LIF received funding from UNAM-PAPIIT IN204224.”

“Co-author DER is grateful to Posgrado en Ciencias del Mar y Limnología, UNAM, for the studies aimed to obtaining the degree of doctor. The authors are grateful to the funding entities (DER received a national graduate studies scholarship from CONAHCYT, Mexico as part of his Doctoral degree research in the Posgrado en Ciencias del Mar y Limnologia. AdLL received a postgraduate scholarship “Estancias Postdoctorales por Mexico” from CONAHCYT. LIF received funding from UNAM-PAPIIT IN204224). The authors are grateful to Dr. Osiris Gaona (IE, UNAM) and Dr. Joanna Ortiz (FC, UNAM) for their technical assistance.”

“DER received a national graduate studies scholarship from CONAHCYT, Mexico as part of his Doctoral degree research in the Posgrado en Ciencias del Mar y Limnologia. AdLL received a postgraduate scholarship “Estancias Postdoctorales por Mexico” from CONAHCYT. LIF received funding from UNAM-PAPIIT IN204224.”

7. We note that Figure 1 in your submission contain [map/satellite] images which may be copyrighted. All PLOS content is published under the Creative Commons Attribution License (CC BY 4.0), which means that the manuscript, images, and Supporting Information files will be freely available online, and any third party is permitted to access, download, copy, distribute, and use these materials in any way, even commercially, with proper attribution. For these reasons, we cannot publish previously copyrighted maps or satellite images created using proprietary data, such as Google software (Google Maps, Street View, and Earth). For more information, see our copyright guidelines: http://journals.plos.org/plosone/s/licenses-and-copyright.

Reviewers' comments:

Reviewer's Responses to Questions

**Comments to the Author**

1. Is the manuscript technically sound, and do the data support the conclusions?

Reviewer #1: Partly

Reviewer #2: Yes

2. Has the statistical analysis been performed appropriately and rigorously? 

Reviewer #1: Yes

Reviewer #2: Yes

3. Have the authors made all data underlying the findings in their manuscript fully available?

Reviewer #1: Yes

Reviewer #2: Yes

4. Is the manuscript presented in an intelligible fashion and written in standard English?

Reviewer #1: Yes

Reviewer #2: No

5. Review Comments to the Author

Reviewer #1: The aim of this study was to describe the composition of bacteria and archaea from mangrove sediments in the Yucatan peninsula coast. Conserved mangroves showed the highest diversity and restored mangroves recover their microbial diversity from the degraded state. This study can help set methodologies that include the microbial component in health assessment and restoration strategies of mangrove forests. Two suggestions for consideration.

1. In materials and methods, it is recommended to define protected, degraded and restored mangroves. It also points out the habitat differences between protected, degraded and restored mangroves.

2. It is necessary to indicate whether the tides are different at the four sampling sites.

Reviewer #2: In this study, results show that microbial diversity in sediments can be used as an index to indicate the environmental health in mangroves. I appreciate the effort to carry out studies that denote new ecological bioindicators, since it is a way to visualise the “ecological future, sustainably through the biota”. However, in my opinion, the purpose and the relevance of this specific study is still unclear. I hope that my comments are useful for the improvement of the article.

Major concerns

1. The construct of the manuscript is poor, right from abstract, introduction to discussion. The ultimate purpose or objectives are missing with some gap of recent citations in the concerned subject.

2. Secondly the language requires native proofreading to avoid phrases and sentences that get repetitive.

3. Please revise the text such that it summarises major findings with significance levels and contextualises their importance within the field, especially in the introduction and discussion.

4. Abstract- Abstract is the briefing that has the merit to stand alone reflecting the purpose of the study, the method, and the findings with a conclusive base. Here descriptive parts of the peninsula are more a fit for the introduction rather to be placed in the abstract.

In here, you bring out the major significant findings, relevance of such studies, etc.

5. The results that are to be highlighted in the abstract needs precision.

6. Line 28-29- The meaning is unclear, rather some values, and/or some specificity of result should be mentioned.

7. The introduction lacks the novelty and significance of the study. Please include the novelty of the approach, the knowledge gaps with recent references and then the significance of such studies that can introduce new proxies for environmental assessors.

8. The Fig.1 needs major improvement especially on the size of the area, the demarcations, and the quality. Please reconstruct a map with a better readability too.

9. Line 90-118: I strongly suggest the incorporation of a table, comprehensively showing the study sites and its details, since it is very confusing at the moment.

10. In Fig. 2, the station SISAL shows higher microbial diversity in degraded area. Please discuss and/or explain why?

11. Line 201- When you say higher average abundance, how high? Mention values of the abundance with the demarcation whether they are significantly higher or not.

12. Line 203- Same as above. Kindly mention how much greater and/or values and whether statistically significant or not.

13. Line 205- Same as above.

14. It is actually applicable to the entire MS that you reconstruct the results section, giving relevant values, significance levels and comparative contexts within experimental groups.

15. In discussion, please highlight the relevance of your data and as to what leads to such results. Furthermore, a comprehensive note on the ecological relevance considering the habitat as a system can substantially help readers understand the significance of such studies.

14. I also suggest changing the titles in the result and discussions. It should be concise yet a congruous heading.

15. Major restructuring and again language correction are suggested for the entire MS.

6. PLOS authors have the option to publish the peer review history of their article (what does this mean?). If published, this will include your full peer review and any attached files.

Reviewer #1: No

Reviewer #2: No

---

## [Author Response · Author response to Decision Letter 0]

6 Jun 2024

Attending the reviewers’ comments to the authors, the following actions were taken:

Reviewer 1 suggestions:

1. In materials and methods, it is recommended to define protected, degraded, and restored mangroves. It also points out the habitat differences between protected, degraded, and restored mangroves. 

2. It is necessary to indicate whether the tides are different at the four sampling sites. 

We appreciate your comments, the previous version was lacking specific descriptions of the sampling sites, and the key characteristics that could affect the microbial composition were not stated explicitly. To clarify these points, we added details that describe the sampling sites on lines 101-114, along with the important monitoring reference from “Laboratorio de Produccion Primaria – CINVESTAV (Herrera-Silveira et al., 2018)”. We included the raw physicochemical information associated with the sediment samples (S1 Table), and critical characteristics of the sampling sites including the mean flooding level, where is evident that there are no differences in tides between the sites (Table 1). Average values of the physicochemical data were also calculated to better understanding of the parameters influencing sampling sites in each conservation status (Table 2). Dominant mangrove species was not the same among locations and conservation status, along with the tables and methods description, the relevance of each species is stated throughout the new version of the text.

 

Reviewer 2 suggestions:

1. In this study, results show that microbial diversity in sediments can be used as an index to indicate the environmental health in mangroves. I appreciate the effort to carry out studies that denote new ecological bioindicators, since it is a way to visualize the “ecological future, sustainably through the biota”. However, in my opinion, the purpose and the relevance of this specific study is still unclear. I hope that my comments are useful for the improvement of the article.

We appreciate the comments, and addressed the observations concerning the purpose, novelty and significance of the study; please refer to the introduction on lines 72-77 and 83-96, where we state the relevance of identifying potential indicator consortia as environmental assessors of the ecosystem. Furthermore, in the discussion section “Potential of microbes to monitor restoration”, we discuss the relevance of specific taxa in ecosystem functionality (see lines 426-454). Describing this range of microbes and its associations with specific conservation conditions highlights the value of a microbial baseline in hydrological restoration scenarios.

To attend major concerns, the following actions were taken:

1. To address the lack of clarity in our previous version we rewrote and restructured the manuscript. As stated in the last suggestion response we clarified the purpose, novelty, and significance of the study (see lines 72-77; 83-96). We also incorporated updated information and recent citations that help to understand the current state of knowledge of the subject. Throughout the manuscript, special attention was taken to the potential drivers of variation of the microbial composition, mangrove species, physicochemical variables and hydrological traits.

2. To address language adjustment and avoidance of redundancy, we asked a native professional to proofread the entire manuscript; we changed the wording, taking care to avoid repetitive sentences.

3. We appreciate your concern; we made several changes mainly to the introduction and discussion sections, including sentences that contextualize the importance within the field, for example, in lines 58-61, “Furthermore, there is little understanding of microorganisms' response and specific role in maintaining conserved mangroves, supporting restoration processes, and enduring degradation since most studies aim to understand local particular processes that influence microbial patterns.” In lines 72-91 “The mangroves in the YP have been degraded because of land use changes, infrastructure, urban expansion, pollution, and extreme climatic events. Many of these impacts have been identified and successful rehabilitation programs have been designed for this region. Although knowledge of microbial component variation associated to different conditions of mangrove forests in the YP has only started to be incorporated in local projects. Work in the Red Sea and in Colombia suggests differences in microbial composition in mangroves with higher anthropic impact than in pristine ecosystems, generally showing higher abundance of microbial taxa that potentially respond to disturbance in degraded areas. Further, studies in China highlight that mangrove restoration stimulates microbial diversity recovery in sediments, nutrient availability and extracellular polymeric substance production. A recent study of the sediment microbial component in Celestun (northwest YP), reported that the red mangrove, Rhizophora mangle, in different ecological types had a more cosmopolitan and heterogenous composition than the less abundant black mangrove, Avicennia germinans, that hosts a specific microbiome, suggesting the relevance of incorporating microbial ecology studies to better define restoration strategies of mangrove forests. Hence, the aim of this study was to make a description of the composition of bacteria and archaea from mangrove sediments along the northern YP coast incorporating mangroves in different status: conserved, degraded and restored.”

4. We appreciate your suggestion. We changed the description of the sampling sites and their characteristics to the methodology section and synthesized site characteristics in Tables 1 and 2.

5. Thanks for your accurate comment; we modified the abstract including specific relative abundance values of major phyla found, we wrote on lines 28-31 “Results showed that although each sampling site had a differentiated microbial composition, the taxa were mainly from Pseudomonadota (13.2 – 23.6%), Desulfobacterota (7.6 - 8.3%), and Chloroflexi (9 – 15.7%) phyla, these were similar between rainy and dry seasons”. We also incorporated the significance values for the alpha diversity conservation status comparisons (see lines 32-34) and stated that some ASVs had significant abundance differences when degraded and conserved samples were compared (see lines 36-38). In this later case, significance values were not explicit since several ASVs are representing the differences within the mentioned taxonomical groups; significance values are referenced in the results section as supplementary tables of the differential abundance analysis (S5 Table – S13 Table).

6. We added specificity to the results stated in the abstract, as exemplified on the previous point to avoid ambiguity. 

7. Thank you for the accurate observation, we modified the introduction and made sure novelty and significance were clearly stated, as mentioned in the response to major concern number 3.

8. Fig 1 was reconstructed, and areas were adjusted correctly, making sure that different vectorial data were at the same scale (1:50000), and the demarcations of coordinates in specific sites were added. The figure’s general quality was enhanced, including a layer that represents the mangrove vegetation coverage.

9. Thanks for the comment. We replaced lines 90-118 and introduced Tables 1 and 2, which mention each sampling site and its characteristics.

10. Thanks for the accurate observation. Considering the more detailed site description and background research from the area, we discussed the possible explanations for the diversity values found in Sisal. Specifically, we highlight that in the degraded plots of Sisal the alpha diversity result was exceptional (see lines 207-209), and we discussed how hydrologic flux linked to seawater and other characteristics may explain this result (see lines 371-384).

11. 12. and 13. Values of the abundances of the reported taxonomical groups were included. Taxa reported with higher abundance in the results section “Sediment microbial diversity and composition” were discussed in comparison to reference conditions (See lines 214-226). The significance values of the taxa that represented differences in contrasting conservation status are referenced in the section “Microbial differential abundance”, where more specific results of these analysis were included in supplementary tables (S5 Table – S13 Table).

14. Results section was reconstructed, including its subtitles. We gave the specific abundance values and included in text and supplementary information that show the significance level of the results, particularly in the comparative context of the different conservation status.

15. The discussion section was rewritten and restructured. The new version highlights the ecological relevance of our results and the variables that explain them; for example, in lines 442-457 we discuss the importance of the effect of the vegetation on microbial composition, as well as the effect of hydrological restoration in ecosystemic succession dynamics. The complete discussion section also gives details of the contexts where our findings can impact the most.

---

## [Decision Letter · Decision Letter 1]

16 Jul 2024

Do restoration strategies in mangroves recover microbial diversity? A case study in the Yucatan peninsula

PONE-D-24-03277R1

Dear Dr. Falcón,

We’re pleased to inform you that your manuscript has been judged scientifically suitable for publication and will be formally accepted for publication once it meets all outstanding technical requirements.

Kind regards,

Jiang-Shiou Hwang, Ph.D.

Academic Editor

PLOS ONE

Additional Editor Comments (optional):

Reviewers' comments:

Reviewer's Responses to Questions

**Comments to the Author**

1. If the authors have adequately addressed your comments raised in a previous round of review and you feel that this manuscript is now acceptable for publication, you may indicate that here to bypass the “Comments to the Author” section, enter your conflict of interest statement in the “Confidential to Editor” section, and submit your "Accept" recommendation.

Reviewer #1: All comments have been addressed

Reviewer #2: All comments have been addressed

2. Is the manuscript technically sound, and do the data support the conclusions?

Reviewer #1: Yes

Reviewer #2: Yes

3. Has the statistical analysis been performed appropriately and rigorously? 

Reviewer #1: Yes

Reviewer #2: Yes

4. Have the authors made all data underlying the findings in their manuscript fully available?

Reviewer #1: Yes

Reviewer #2: Yes

5. Is the manuscript presented in an intelligible fashion and written in standard English?

Reviewer #1: Yes

Reviewer #2: Yes

6. Review Comments to the Author

Reviewer #1: The author has modified and responded to the comments of the reviewers, meeting the requirements for publication. It is suggested to add "Mexico" at the end of the title, that is “Do restoration strategies in mangroves recover microbial diversity? A case study in the Yucatan peninsula, Mexico”.

Reviewer #2: The authors have made necessary corrections and have satisfactorily addressed all the raised concerns. Now the paper is ready to be accepted for publication.

7. PLOS authors have the option to publish the peer review history of their article (what does this mean?). If published, this will include your full peer review and any attached files.

Reviewer #1: No

Reviewer #2: **Yes: **Shagnika Das

---

## [Editor Report · Acceptance letter]

7 Aug 2024

PONE-D-24-03277R1 

PLOS ONE

Dear Dr. Falcón, 

I'm pleased to inform you that your manuscript has been deemed suitable for publication in PLOS ONE. Congratulations! Your manuscript is now being handed over to our production team.

Kind regards, 

on behalf of

Prof. Jiang-Shiou Hwang 

Academic Editor

PLOS ONE